# Pathway-guided monitoring of the disease course in bladder cancer with longitudinal urine proteomics

Luís Botelho Carvalho [1,2], José Luis Capelo [1,2], Carlos Lodeiro [1,2], Rajiv Dhir[3], Luis Campos Pinheiro[4,5], Hugo López-Fernández[6,7], Gonçalo Martins[1,2], Mariana Medeiros[4,5], Fernando Díaz[8] & Hugo Miguel Santos [1,2,3✉]

## Abstract

**Background** Monitoring bladder cancer over time requires invasive and costly procedures. Less invasive approaches are required using readily available biological samples such as urine. In this study, we demonstrate a method for longitudinal analysis of the urine proteome to monitor the disease course in patients with bladder cancer.

**Methods** We compared the urine proteomes of patients who experienced recurrence and/or progression ($n = 13$) with those who did not ($n = 17$). We identified differentially expressed proteins within various pathways related to the hallmarks of cancer. The variation of such pathways during the disease course was determined using our differential personal pathway index (dPPi) calculation, which could indicate disease progression and the need for medical intervention.

**Results** Seven hallmark pathways are used to develop the dPPi. We demonstrate that we can successfully longitudinally monitor the disease course in bladder cancer patients through a combination of urine proteomic analysis and the dPPi calculation, over a period of 62 months.

**Conclusions** Using the information contained in the patient's urinary proteome, the dPPi reflects the individual's course of bladder cancer, and helps to optimise the use of more invasive procedures such as cystoscopy.

## Plain language summary

Bladder cancer must be closely monitored for progression, but this requires expensive and invasive procedures such as cystoscopy. Less invasive procedures using readily available samples such as urine are needed. Here, we present an approach that measures the levels of various proteins in the urine. We compare protein levels at different points during the disease course in patients with bladder cancer, and show this helps to flag disease recurrence and the need for medical intervention. Our approach could help clinicians to determine which patients require more invasive testing and treatment.

[1] BIOSCOPE Research Group, LAQV-REQUIMTE, Department of Chemistry, NOVA School of Science and Technology, Universidade NOVA de Lisboa, 2829-516 Caparica, Portugal. [2] PROTEOMASS Scientific Society, Madan Parque, Rua dos Inventores, 2825-182 Caparica, Portugal. [3] Department of Pathology, University of Pittsburgh Medical Center, Pittsburgh, PA, USA. [4] Urology Department, Central Lisbon Hospital Center, Lisbon, Portugal. [5] NOVA Medical School, NOVA University of Lisbon, Lisbon, Portugal. [6] CINBIO, Universidade de Vigo, Department of Computer Science, ESEI-Escuela Superior de Ingeniería Informática, 32004 Ourense, Spain. [7] SING Research Group, Galicia Sur Health Research Institute (IIS Galicia Sur), SERGAS-UVIGO, 36213 Vigo, Spain. [8] Department of Computer Science, Universidad de Valladolid, Escuela de Ingeniería Informática, 40005 Segovia, Spain. ✉email: hmsantos@fct.unl.pt

Bladder cancer (BC) is among the worst neoplasms because of its high incidence and mortality[1,2] and because cystoscopy is necessary to diagnose BC. If BC is confirmed, the patient undergoes surgery, then must undergo a control cystoscopy approximately every 3 months[3,4]. Cystoscopy is an expensive intervention that requires surgical facilities and a minimum of two physicians and one nurse. However, the procedure can also have a detrimental effect on the patient's state of mind, as cystoscopy is highly invasive and is performed through the urethra. Furthermore, some patients experience a rapid recurrence; thus, the control cystoscopy is performed too late, and transurethral resection of the bladder tumour is needed. Therefore, new approaches to diagnose and monitor BC are needed.

The biochemical changes that occur at the protein level in an individual after they develop a disease provides that can enable physicians to adjust therapy, provide follow up medical care and personalise their care[5,6]. The concept of performing longitudinal profiling with individual proteomes has been described in the literature;[7–9] however, to date, there are few easy ways to achieve this in a simple, straightforward and robust manner. To our knowledge, the possibility of monitoring the course of BC over time using the urinary proteome has not yet been described in the literature. To overcome this gap, we explored changes in the urinary proteome of BC patients using high-resolution mass spectrometry-based proteomics in conjunction with advanced bioinformatics tools. Thus, we introduced, a concept named the differential personal pathway index (dPPi), which uses the variation in the expression of selected biochemical pathways, which is calculated using a large number of urine proteins that are linked to Hanahan and Weinberg's biological hallmarks of cancer (HWhc[10,11]). Thus, based on variation in the levels of tens of signature urine proteins, the dPPi reflects both the response of the individual's urine proteome to whatever course the disease takes and any medical care that ensues. The variation reflects the status of the biochemical pathways to which the proteins correspond, and these pathways are specifically chosen to combine multiple sources of diagnostic information. Thus, in this work, we showed how to monitor the progression of BC in a comprehensive yet straightforward and personalised manner using the patient's urine proteome. Essentially, a more positive dPPi indicates a potentially more severe clinical course of BC; whereas a more negative dPPi indicates a better clinical outlook. Therefore, the dPPi flags the need for medical intervention at an early stage. This work further demonstrates this concept using six patients diagnosed with T1-stage BC who were followed for up to 62 months.

## Methods

**Patients**. Human mid-stream second void morning urine specimens were collected from patients with a pT1 bladder cancer diagnosis that was confirmed by pathological examination. All patients were informed about the study and signed informed consent according to the policies of the Central Lisbon Hospital Center Ethics Committee, who approved our study. The ethics approval number is 669/2018. The patients that were enroled in the study were selected based on the following criteria: (a) Inclusion—clear bladder cancer diagnosis, and (b) Exclusion—no records of urinary cancer history, no HIV, no organ transplant, and no recent chemo/radiotherapy. We compared the proteome of 30 patients from two groups—those who had experienced BC recurrence and/or progression (a total of 13) and those who had not (a total of 17). Fig. 1a depicts the steps from proteome analysis to pathway, and clinical data interpretation, while the medical information for the cohorts of patients is presented in Fig. 1b. Furthermore, we followed six patients for up to 62 months.

**Sample treatment**. Urine samples were collected in 50 mL centrifuge tubes (DB Falcon) that contained 38 mg of boric acid (Sigma-Aldrich) to prevent bacterial growth[12]. Samples with haematuria were not included in our study. The urine samples were centrifuged at $5000 \times g$ for 20 min to remove cell debris, and 10 mL aliquots of the resulting supernatants were stored at $-60\,°C$ until further use. An aliquot of urine (10 mL) was concentrated by centrifugal ultrafiltration using a VivaSpin 15R (10 kDa MWCO, Sartorious) at $5000 \times g$ for 20 min. Finally, the concentrated urinary proteomes were quantified via a 96 well plate Bradford protein assay (Sigma-Aldrich) using bovine serum albumin to generate the calibration curve.

**Filter-aided sample preparation (FASP)**. Filter Aided Sample Preparation (FASP) was used to perform urinary proteome digestion, following a modified FASP method[5,13–15]. Briefly, different amounts of protein ranging from 50 and 100 μg were diluted with $H_2O$ to obtain 200 μL and an additional 200 μL of 8 M urea (Sigma-Aldrich), 75 mM Tris-HCl (Sigma-Aldrich), 100 mM NaCl (Sigma-Aldrich), and 0.02% SDS (Sigma-Aldrich). Then this protein solution was loaded onto a VivaSpin 500 (10 kDa MWCO, Sartorious) and centrifuged for 20 min at $14,000 \times g$. Each sample was processed in duplicate. The proteins in the membrane were washed with 200 μL of 8 M urea 25 mM AmBic solution and then centrifuged for 20 min at $14,000 \times g$. Protein disulphide bonds were reduced by adding 200 μL of 50 mM dithiothreitol (DTT) in 8 M urea and 25 mM AmBic and incubated for 60 min at 37 °C. Then, centrifugation for 20 min at $14,000 \times g$ was performed, and the sample was alkylated for 45 min in the dark by adding 100 μL 50 mM iodoacetamide in 8 M urea and 25 mM AmBic solution. Subsequently, centrifugation for 20 min at $14,000 \times g$ was performed, and then the samples were washed twice with 200 μL of 25 mM AmBic. Finally, protein digestion was performed by adding 100 μL of trypsin solution (1:30 trypsin to protein ratio) prepared in 12.5 mM AmBic. Protein digestion was performed overnight (~16 h) at 37 °C, and then the peptides were collected by 20 min centrifugation at $14,000 \times g$, followed by two additional membrane washing steps with 100 μL of 3% (v/v) acetonitrile that contained 0.1% (v/v) aqueous formic acid and centrifugation of 20 min at $14,000 \times g$. Finally, peptides were transferred to a 500 μL microtube, dried and stored at $-20\,°C$ until further analysis by Nano-LC-MS/MS.

**LC–MS/MS**. LC-MS /MS analysis was carried out using an Ultimate 3000 nano-LC system coupled to an Impact HD (Bruker Daltonics) with a CaptiveSpray nanoBooster that used acetonitrile as a dopant. The protein digests were resuspended in 100 μL of 3% (v/v) acetonitrile containing 0.1% (v/v) aqueous formic acid (FA) and were sonicated for 10 min using an ultrasonic bath at 100% ultrasonic amplitude, and 35 kHz ultrasonic frequency. Afterwards, the samples were quantified using a Pierce™ quantitative colorimetric peptide assay. Then, 3 μL of protein digest containing 563 ng of peptides was loaded onto a trap column (Acclaim PepMap100, 5 μm, 100 Å, 300 μm i.d. × 5 mm) and desalted for 5 min from 3 to 5% B (B: 90% acetonitrile 0.08% FA) at a flow rate of 15 μL min⁻¹. The peptides were separated using an analytical column (Acclaim™ PepMap™ 100C18, 2 μm, 0.075 mm i.d × 150 mm) with a linear gradient at 300 nL min⁻¹ (mobile phase A: aqueous FA 0.1% (v/v); mobile phase B 90% (v/v) acetonitrile and 0.08% (v/v) FA) 5–90 min from 5 to 35% mobile phase B, 90–100 min linear gradient from 35 to 95% of mobile phase B, 100–110 min 95% B. Chromatographic separation was carried out at 35 °C. MS acquisition was set to cycles of MS (2 Hz), followed by MS/MS (8–32 Hz) with a cycle time of 3.0 s and active exclusion (precursors were excluded from the precursor selection for 0.5 min after the

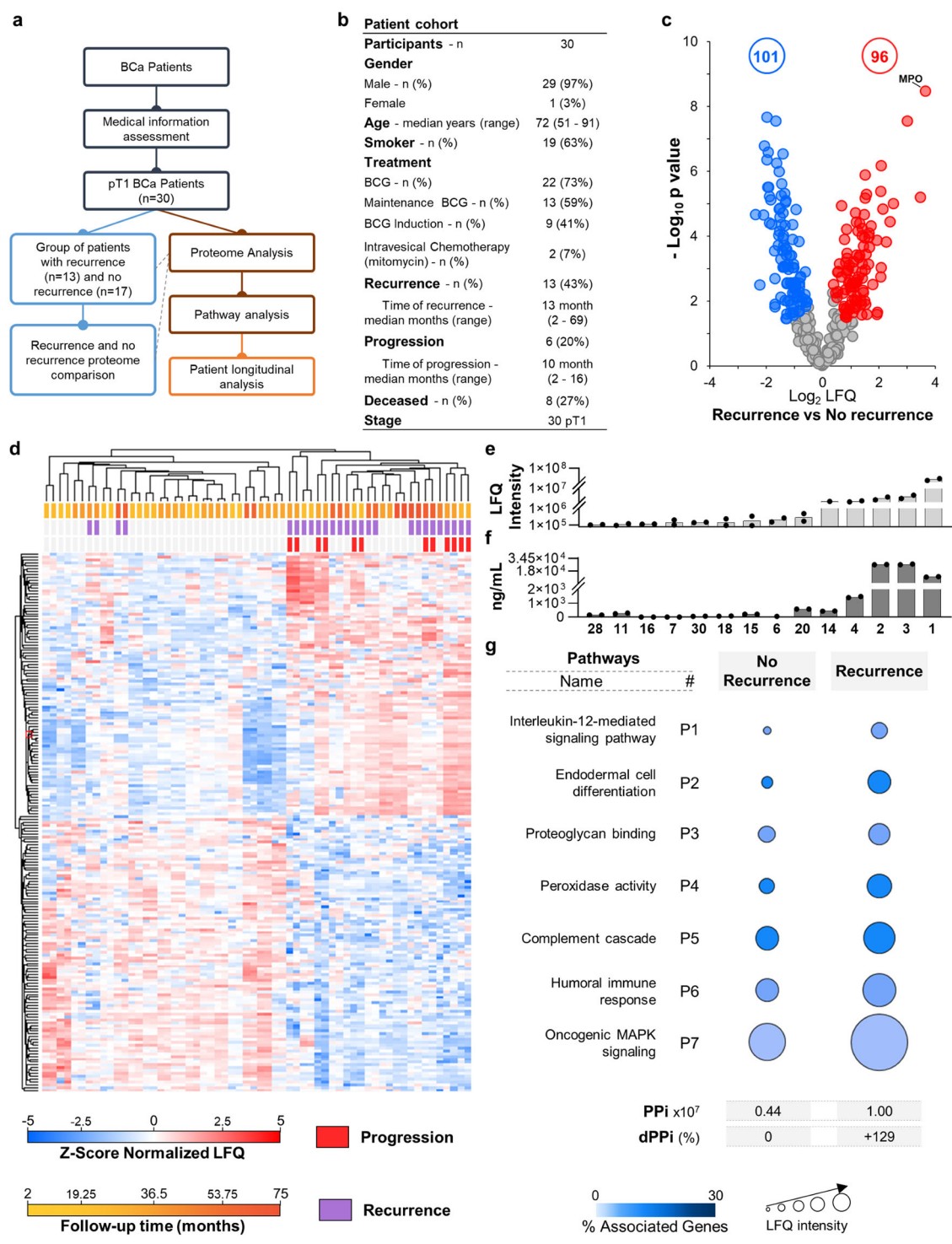

acquisition of the 1 MS/MS spectrum; the intensity threshold for fragmentation was 2500 counts). As, the active exclusion was set to one, we reconsidered the precursor and whether the intensity of a precursor increased by a factor of three; this mass was taken from the temporarily exclusion list and fragmented again, ensuring that fragment spectra were taken near to the peak maximum. All spectra were acquired in the range 150–2200 m/z. The mass spectrometry proteomics data were deposited in the ProteomeXchange Consortium[16] via the PRIDE[17] partner repository with the data set identifier PXD025139. To confirm myeloperoxidase label-free quantification levels, validation was performed using a commercially available ELISA kit (ab136943 - Myeloperoxidase (MPO)

Human ELISA kit from Abcam). The method was carried out according to the standard protocol described by the manufacturer.

**Bioinformatics and statistical analyses**. Relative label-free quantification was performed using the precursor signal intensity method and delayed normalisation, MaxLFQ, on MaxQuant software V2.0.3.0[18]. All raw files were processed in a single run with default parameters[19,20]. Database searches were performed using Andromeda against the human UniProt UP000005640_9606 database (20,600 sequences; 11,395,157 residues, downloaded on April 27 2021)[21]. Searches were configured with cysteine

**Fig. 1 Proteome analysis of 30 pT1 bladder cancer patients. a** Workflow of the experimental design. **b** Clinical information of the patient cohort used in the study. **c** Volcano plots showing statistically significant changes in protein levels in T1-BC patients with recurrences and/or progression ($n = 13$ patients, two technical replicates per patient) compared to the group with nor recurrence ($n = 17$ patients, two technical replicates per patient), according to Student's t test (FDR 0.05 and $S_0$ of 0.1). Dots represent proteins that show statistically significant increases (red), decreases (blue) nonstatistically significant changes (grey). The total numbers of proteins that exhibited statistically significant differential expression are shown in the corresponding-coloured circles. **d** Hierarchical clustering of the T1-BC patients into two groups, as expected, recurrence and no recurrence (average linkage, no constraint, preprocessing with k-means, and Euclidean distance, $n = 30$ patients, two technical replicates per patient). **e, f** Validation of mass spectrometry data using myeloperoxidase (MPO) protein label-free quantification (LFQ) and ELISA (ng/mL). Bars represent the average of two technical replicates. The MPO protein was recently proposed to be a marker of poor prognosis in lung[27] and ovarian[28] carcinomas. To the best of our knowledge, no studies report MPO overexpression as a prognostic tool for BC carcinoma. **g** Pathways selected for calculating the Personal Pathway Index, after applying the guidelines provided by Hanahan and Weinberg[10, 11] to the proteins showing statistically significant differential expression derived from the volcano plot in **c** (Supplementary Data 1). Differentially expressed proteins were grouped according to their functions, and networks were analysed in ClueGo (v 2.5.8), and Cytoscape (v 3.8.2). The LFQ average for all proteins involved in the pathway is expressed as a circle area plot. The maximum number of proteins identified in one single urine of our set was 494. The number of unique proteins considering all the samples of our set was about 754. For our studies, we have chosen the proteins that are found common in at least 50% of the urines of each group. This way, the number of proteins used in Fig. 1 (recurrence versus no recurrence) was 380.

| Pathway Number | GOID | Name | Ontology Source | Associated Proteins Found |
|---|---|---|---|---|
| | | | | |

**Table 1 Pathways and proteins used on the longitudinal follow up of bladder cancer patients.**

| Pathway Number | GOID | Name | Ontology Source | Associated Proteins Found |
|---|---|---|---|---|
| P1 | GO:0035722 | interleukin-12-mediated signalling pathway | GO_BiologicalProcess-EBI-UniProt-GOA-ACAP-ARAP_08.05.2020_00h00 | [LCP1, PPIA, S100A9ᵃ] |
| P2 | GO:0035987 | endodermal cell differentiation | GO_BiologicalProcess-EBI-UniProt-GOA-ACAP-ARAP_08.05.2020_00h00 | [FN1, MMP8, MMP9ᵃ] |
| P3 | GO:0043394 | proteoglycan binding | GO_MolecularFunction-EBI-UniProt-GOA-ACAP-ARAP_08.05.2020_00h00 | [AZU1, CFH, CTSB, FN1, HRG, LCP1, NID1] |
| P4 | GO:0004601 | peroxidase activity | GO_BiologicalProcess-EBI-UniProt-GOA-ACAP-ARAP_08.05.2020_00h00 | [CAT, HBA1, HBB, HBD, MPO] |
| P5 | R-HSA:166658 | Complement cascade | REACTOME_Pathways_08.05.2020 | [C3, C4A, C9, CD55, CFH, F2] |
| P6 | GO:0006959 | humoral immune response | GO_BiologicalProcess-EBI-UniProt-GOA-ACAP-ARAP_08.05.2020_00h00 | [A2M, APCS, AZU1, C3, C4A, C9, CAMP, CD55, CFH, F2, FGBᵃ, HPX, HRG, IGHG1, IGKV2D-30, KRT1, LCN2, LTF, LYZ, PRTN3, S100A9ᵃ] |
| P7 | R-HSA:6802957 | Oncogenic MAPK signalling | REACTOME_Pathways_08.05.2020 | [FGBᵃ, FGGᵃ, FN1] |

ᵃIndicates a protein previously identified as a BC biomarker

carbamidomethylation as a fixed modification and N-terminal acetylation and methionine oxidation as variable modifications. We set the false discovery rate (FDR) to 0.01 for protein and peptide levels with a minimum length of seven amino acids for peptides. The FDR was determined by searching a reverse database. Enzyme specificity was set as C-terminal to arginine and lysine as expected using trypsin. A maximum of two missed cleavages were allowed. Data processing was performed using Perseus (version 1.6.15.0) with default settings[22]. Protein group LFQ intensities were log2-transformed, and the quantitative profiles were filtered for missing values with the following settings: min valid percentage of 50% in at least one group and values greater than 0. To overcome the obstacle introduced by missing LFQ values, the missing values were imputed using the parameters, with = 0.5 and downshift = 1.8. The list of differentially expressed proteins was obtained with a two-tailed Student's t test between the two groups (permutation-based FDR 0.05 and S0 of 0.1), and then the log ratios were calculated as the difference in average log$_2$ LFQ intensity values between the tested conditions. Perseus was also used to obtain clusters, using average linkage, no constraint, preprocess with k-means and Euclidean distance between column trees. GraphPad Prism 8.2.1. was employed for graphic construction and statistical analysis. Cytoscape V3.8.2 and the application ClueGo V2.5.8 was used for pathway enrichment analysis. Reactome and biological process GO terms were used as ontologu databases.

**dPPi calculation**. The dPPi is calculated from the numerical output of proteomics analysis of urine at different time points, as follows:

$$dPPi = \frac{\sum_{nx=1}^{nx=i} Protnx\, LFQ_{ty} - \sum_{nx=1}^{nx=i} Protnx\, LFQ_{t1}}{\sum_{nx=1}^{nx=i} Protnx\, LFQ_{t1}} \quad \text{(I)}$$

where $\sum_{nx=1}^{nx=i} Protnx\, LFQ_{ty}$ accounts for the sum of all label-free quantitation (LFQ) values of proteins of interest at a given time $t_y$, while $\sum_{nx=1}^{nx=i} Protnx\, LFQ_{t1}$ accounts for all LFQ measurements of the same proteins at the time of disease onset $t_1$.

**Reporting summary**. Further information on research design is available in the Nature Portfolio Reporting Summary linked to this article.

## Results and discussion

**Selection of pathway hallmarks of cancer**. The differentially expressed proteins between the two groups ($n = 197$, Fig. 1c and

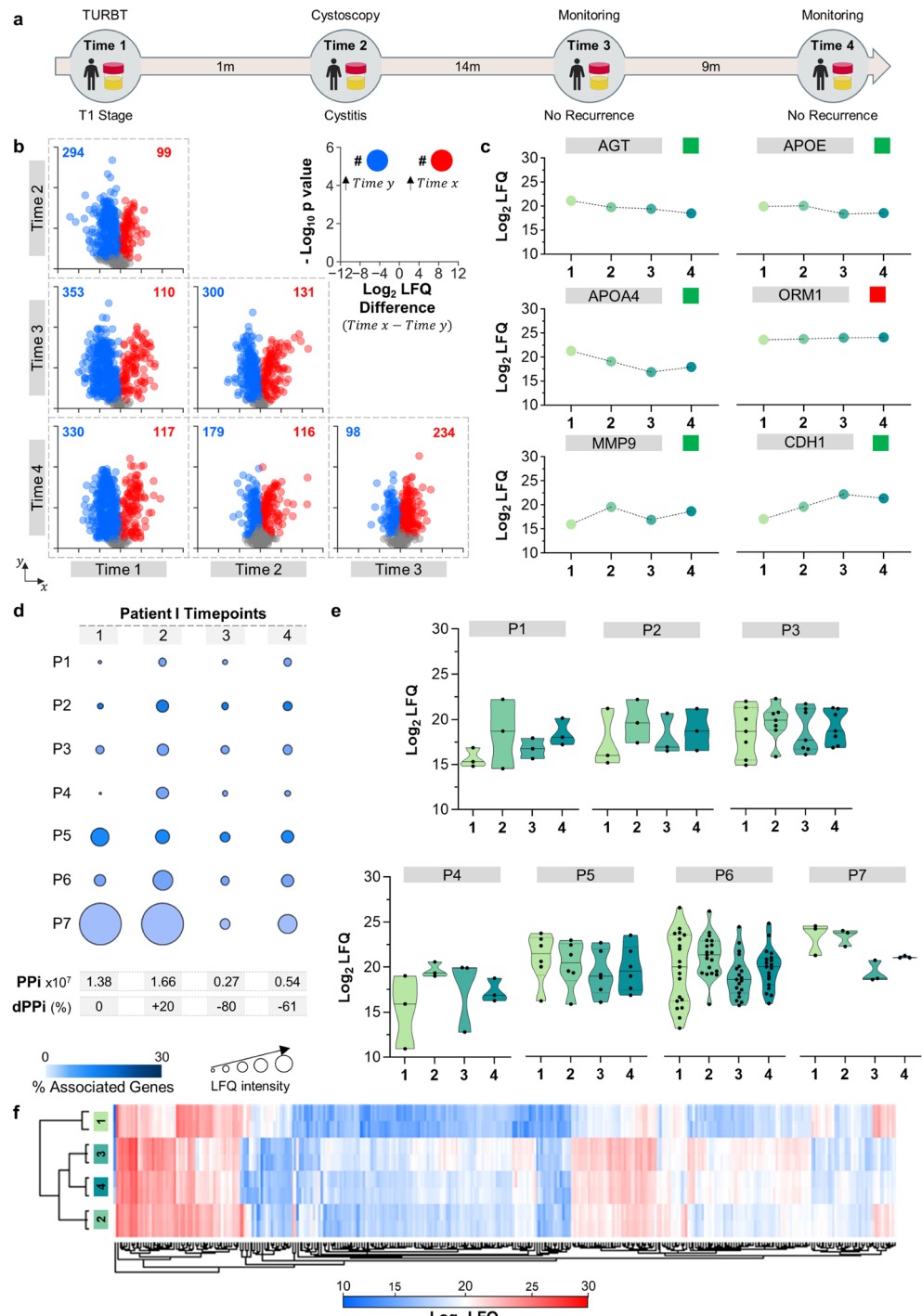

**Fig. 2 Longitudinal analysis of the urinary proteome of patient I. a** Timeline of the disease course in Patient I shows the intervals in months (m) between medical interventions and urine timepoint 1–4. TURBT: transurethral resection of bladder tumour; Re-TURBT: repeated transurethral resection of bladder tumour. **b** Volcano plot matrix showing proteome changes in the urine samples at time points 1–4. The dots represent the proteins that showed a statistically significant increase (red), decrease (blue) or nonstatistically significant changes (grey) according to the Student's t test (FDR 0.05 and S0 of 0.1). **c** Variation in the following known protein biomarkers for bladder cancer: angiotensinogen (AGT)[35], apolipoprotein E (APOE)[36], matrix metalloproteinase-9 (MMP9)[37], apolipoprotein A-IV (APOA4)[29], alpha-1-acid glycoprotein 1 (ORM1)[38] and cadherin-1 (CDH1)[30, 31]. The green and red squares indicate whether the variation in the urine of Patient I matched or not, respectively, with trends reported for each marker in the literature. Dots represent the average of two biological samples with two technical replicates each. **d** Personal pathway index (PPi) at each time point and estimated differential PPI (dPPi) were calculated as explained in the text. P1: interleukin-12-mediated signalling pathway; P2: endodermal cell differentiation; P3, proteoglycan binding; Pa: peroxidase activity; P5: complement cascade; P6: humoral immune response; P7: oncogenic MAPK signalling. **e** Distribution and density variation protein LFQ values (including two biological replicates) at each timepoint point for each pathway assessed. Continuous bar in the middle represents the median. The thin discontinuous line represents the quartile lines. **f** Hierarchical clustering of the three urinary proteomes of Patient I. Protein LFQ values were used to perform the cluster analysis (with average linkage, no constraint, preprocessing with k-means and Euclidean distance).

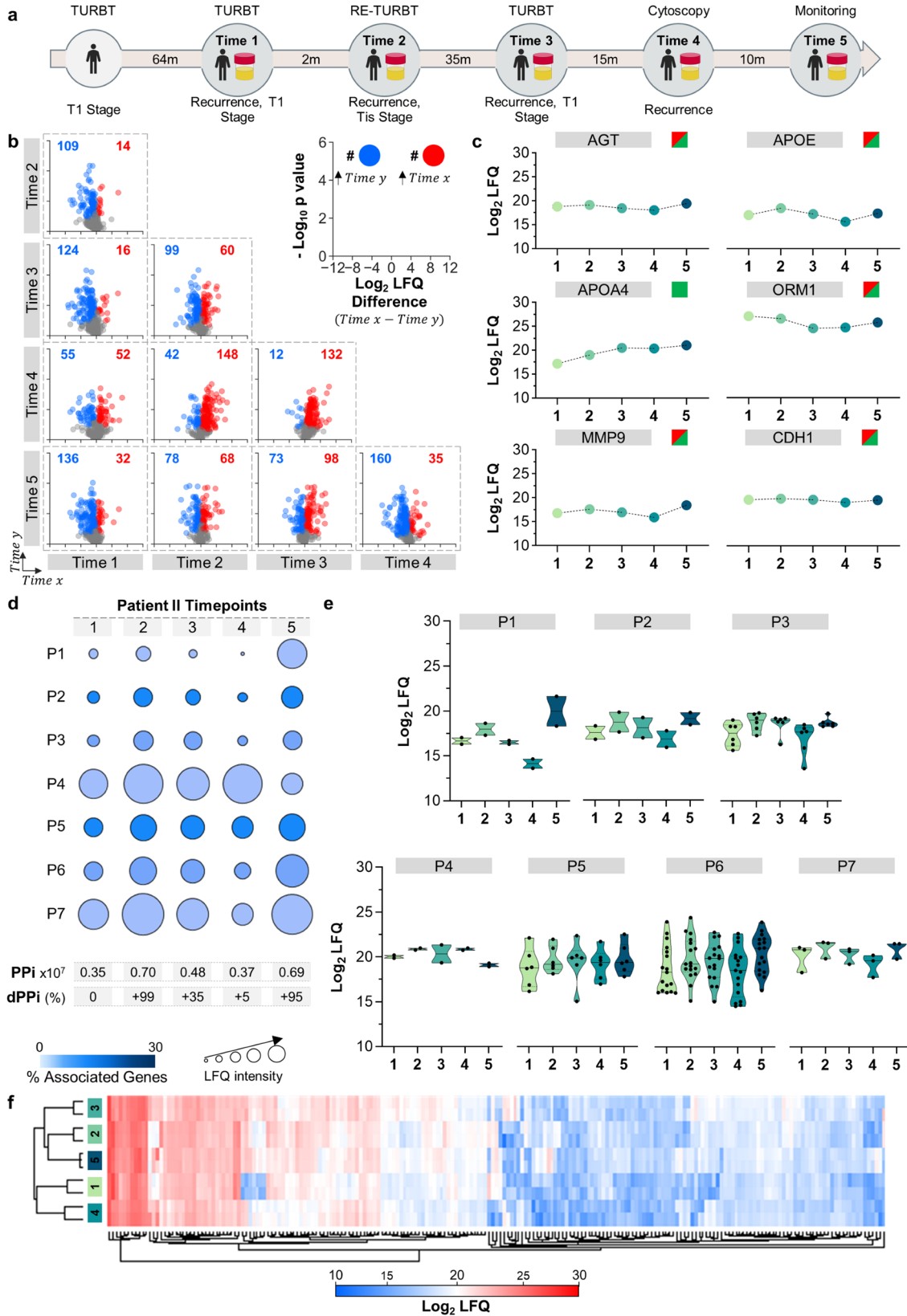

Supplementary Data 1) were submitted to clue-go analysis (cystoscope) to select the biochemical pathways involved in recurrence and/or progression. The 197 differentially expressed proteins distinguished 93.5% of patients with recurrence and 100% of patients with progression, as shown in Fig. 1d. The statistically significant pathways were cross-checked against the

HWhc[10,11], and seven pathways were selected. These pathways involve 35 related proteins (Fig. 1g. and Table 1). These 35 proteins were used to calculate the dPPi as defined in Eq. (I). Four of these 35 proteins have been described in literature as potential biomarkers for bladder cancer (indicated with ᵃ in Table 1)[23–25]. Two of these four proteins namely fibrinogen β chain[25,26] and

**Fig. 3 Longitudinal analysis of the urinary proteome of patient II. a** Timeline of disease course in Patient II showing the intervals in months (m) between medical interventions and urine timepoint 1 to 5. TURBT: transurethral resection of bladder tumour; Re-TURBT: repeated transurethral resection of bladder tumour. **b** Volcano plot matrix showing proteome changes in urine collected at time points 1–5. The dots represent proteins showing statistically significant increases (red), decreases (blue) or nonstatistically significant changes (grey) according to the Student's $t$ test (FDR 0.05 and S0 of 0.1). **c** Variation in the known protein biomarkers for bladder cancer, including angiotensinogen (AGT)[35], apolipoprotein E (APOE)[36], matrix metalloproteinase-9 (MMP9)[37], apolipoprotein A-IV (APOA4)[29], alpha-1-acid glycoprotein 1 (ORM1)[38] and cadherin-1 (CDH1)[30, 31]. The green and red squares indicate whether the variation in the urine of Patient II matched or not, respectively, with trends reported for each marker in the literature. Dots represent the average of two biological samples with two technical replicates each. **d** Personal pathway index (PPi) at each time point and the estimated differential PPi (dPPi) were calculated as explained in the text. P1: interleukin-12-mediated signalling pathway; P2: endodermal cell differentiation; P3, proteoglycan binding; P4: peroxidase activity; P5: complement cascade; P6: humoral immune response; P7: oncogenic MAPK signalling. **e** Distribution and density variation protein LFQ values (including two biological replicates) at each timepoint for each pathway assessed. Continuous bar in the middle represents the median. The thin discontinuous line represents the quartile lines. **f** Hierarchical clustering of the four urinary proteomes for Patient II. The protein LFQ values were used to perform the cluster analysis (with average linkage, no constraint, preprocessing with k-means and Euclidean distance).

fibrinogen γ chain[25] have been described as having increased expression in urine of BC patients. In addition, we found the protein S100A9 which has been described as a biomarker candidate in plasma of BC patients[23]. Protein myeloperoxidase (MPO) was found to exhibit the greatest increases in levels in patients with recurrence (Fig. 1c). MPO has been recently proposed to be a marker of poor prognosis in lung[27] and ovarian[28] carcinomas. Therefore, MPO was used for quality control to validate the mass spectrometry data using ELISA as an orthogonal method (Fig. 1e, f). Next, we investigated how the status of the seven pathways changed over time by following six patients with BC for up to 40 months. A comprehensive flow chart of the process is provided in Supplementary Fig. 1.

**Longitudinal follow-up of BC patients.** We followed six patients longitudinally over the course of their diagnosis and treatment. In this section, we will discuss three of the six patients in detail, and the other three are displayed in Supplementary Figs. 3, 4 and 5.

Patient I presented with symptoms that matched BC and was subsequently diagnosed. One reference urine sample was taken before urinary and bladder exploration was performed (see Fig. 2a for a comprehensive analysis timeline). Then, the urine proteome was extracted and analysed via mass spectrometry, and the PPi was calculated as explained in Eq. (I) and is presented in Fig. 2d. Physicians diagnosed this patient with T1-stage BC and performed a transurethral resection of the bladder tumour, TURBT. Then, one month later, a second analysis of the urinary proteome revealed a difference of +35% between the PPi at time point 1 and at time point 2 (Fig. 2d, PPi2 – PPi1), suggesting that the patient had not improved. In fact, all the hallmark pathways were obtained with higher values. This prompted us to call the patient and perform a medical inspection of the bladder, and the patient was diagnosed with cystitis; thus, no surgical intervention was needed. Another advantage of this type of analysis is that the mass spec LFQ values of individual protein biomarkers can also be used to obtain additional information about the course of the disease (Fig. 2c). For instance, APOA4, a urinary candidate biomarker for the presence of BC, exhibited lower value at timepoint 2 which might be a good indicator in terms of cancer progression. Increased levels of APOA4 in urine have been linked to bladder cancer[29]. The same applies to the biomarker CDH1, in which an increment might also be a good signal[30,31]. On the other hand, MMP9 is a biomarker linked to inflammation, and MMP9 exhibited an augmented value at timepoint 2, which is a negative indicator that is not necessarily related to cancer[32].

In summary, the changes in these biomarkers are consistent with cystitis rather than a recurrence or progression. Further information supporting this is that the complex MMP9/NGAL has been reported as increased in the urine of children with acute cystitis[32]. At timepoint 2, both proteins are augmented. This result shows that

dPPi is useful as an alert tool to follow individual variations in the urine proteome caused by disease. Another cystoscopy three months later was performed and revealed no evidence of recurrence. Fourteen months later, a third urine sample was monitored again, showing a dPPi of −80% relative to time 1 (Fig. 2d timepoint 3), indicating potentially important immunological and inflammatory responses were downregulated (Fig. 2e), reflecting an improvement in the patient's condition, as confirmed by medical analysis. Nine months later, a fourth urine analysis revealed an increase in the dPPi index from −80% to −61%, so we concluded that no intervention was necessary. This was confirmed by the physicians. However, there is a clear increase in the MAPK oncogenic signalling pathway. This pathway has been reported to be involved in BC development[33], and overexpression of MAPK signalling proteins has been linked to a mutation in the P53 gene and poor prognosis[34]. Furthermore, the variations in APOA4 and CDH1 indicated that the cancer progression had potentially worsened. Additionally, a change in MMP9 indicated a potential inflammatory response. Altogether, it was concluded that the patient was in a possible initial stage of recurrence. We recommended following this patient more frequently to check the dPPi levels in urine..

Patient II is an interesting case due to their frequent recurrences (Fig. 3). The first proteome analysis of the urine was performed 64 months after the onset of BC. Then, the PPi for this urine, which is presented in Fig. 3d timepoint 1, was calculated. At this point, the patient had a TURBT intervention. Two months later, a new urine analysis revealed a dPPi variation of +99%, (Fig. 3d timepoint 2), indicating that the patient's course was worsening. It is worth noting the increase in the expression of the MAPK pathway (P7, Fig. 3d timepoint 2). Based on these data, the patient underwent a urological assessment that revealed another tumour recurrence, confirming that the alert from the dPPi approach is valuable. Another TURBT was performed, and BC was documented.

After 35 months, a routine surveillance assessment was performed, and the urine proteome analysis revealed a dPPi of +35% relative to the time one reference value. The patient thus underwent another cystoscopy evaluation, which agreed with the dPPi alert and demonstrated the recurrence of BC. Consequently, a new TURBT was performed. A fourth urine proteomic analysis was completed 15 months later (Fig. 3d timepoint 4), which showed a change in the dPPi value of +5%, suggesting potential disease progression. Our prediction was again confirmed by cystoscopy as recurrence was recorded. Finally, ten months later an increase of +95% in the dPPi suggested that the disease was worsening. According to our medical intervention advice, the patient underwent one cystoscopy that confirmed our result. The patient was submitted to a TURBT.

Patient III (Fig. 4) presented with symptoms of BC, and a TURBT was performed because the cystoscopy results were positive. Histopathological analysis documented stage T1 BC. The patient's

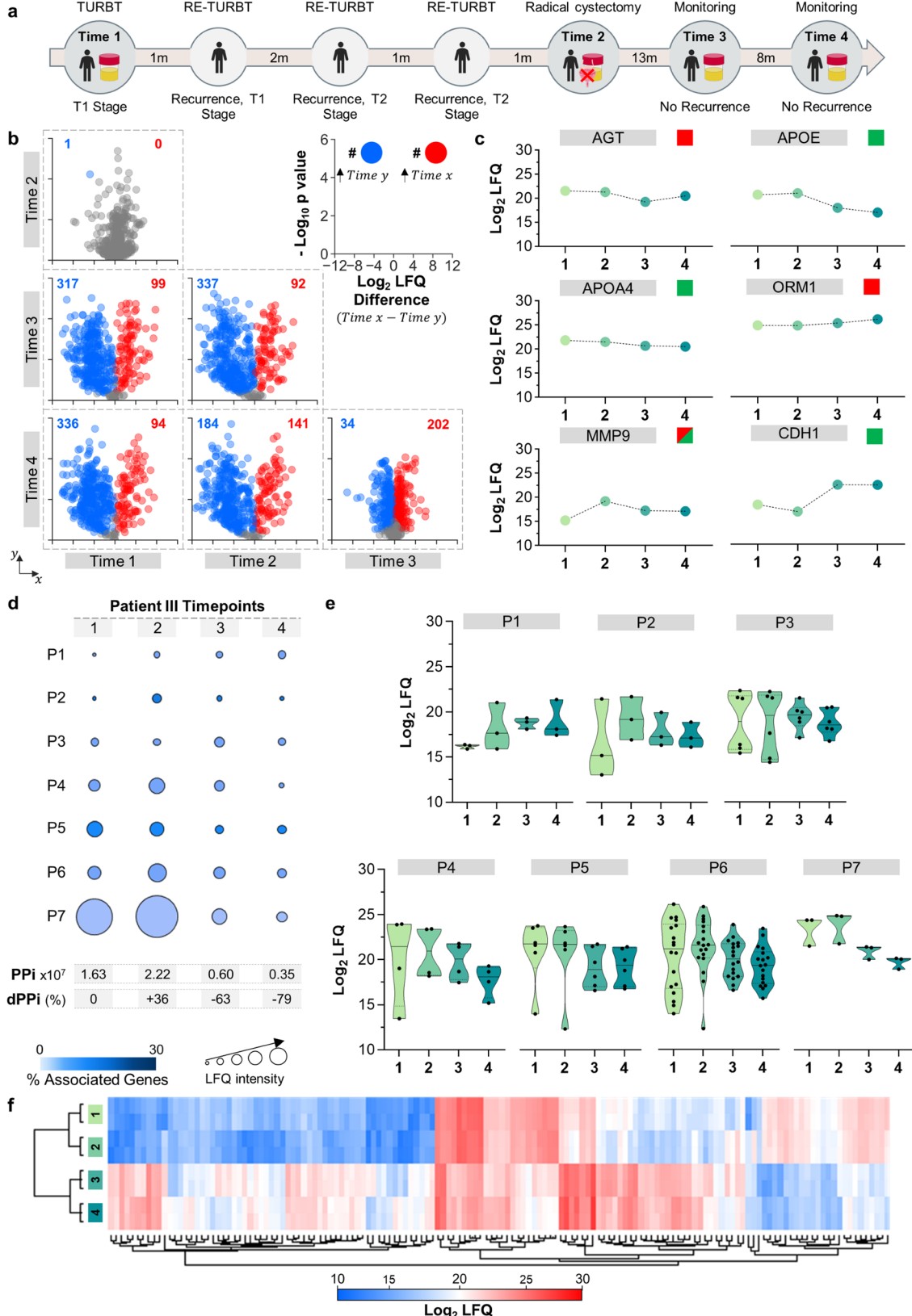

urine sample was collected before medical intervention, and the dPPi was calculated, as shown in Fig. 4d timepoint 1. Afterwards, the patient underwent three more TURBT procedures within the following four months. A second urine sample was taken one month after the fourth TURBT was performed (time point 2). The dPPi augmented a +36% (Fig. 4d timepoint 2), presenting seven

pathways that were increased. Again, the value of the dPPi approach was illustrated by the variation observed in the MAPK signalling pathway (P7 in Fig. 4d).

Our proteomics-based alert was confirmed as the medical exploration indicated that the disease had progressed to the T2-stage, and a radical cystectomy was perfomed. The urine

**Fig. 4 Longitudinal analysis of the urinary proteome of patient III. a** Timeline of the disease course in Patient III showing intervals in months (m) between medical interventions and urine timepoints 1 to 4. TURBT: transurethral resection of bladder tumour; Re-TURBT: repeated transurethral resection of bladder tumour. **b** Volcano plot matrix showing proteome changes in urine collected at time points 1–4. The dots represent proteins showing statistically significant increases (red), decreases (blue) or nonstatistically significant changes (grey) at the different time points according to Student's $t$ test (FDR 0.05 and S0 of 0.1). **c** Variation in the following known protein biomarkers for bladder cancer: angiotensinogen (AGT)[35], apolipoprotein E (APOE)[36], matrix metalloproteinase-9 (MMP9)[37], apolipoprotein A-IV (APOA4)[29], alpha-1-acid glycoprotein 1 (ORM1)[38] and cadherin-1 (CDH1)[30, 31]. The green and red squares indicate whether the variation in the urine of Patient III matched or not, respectively, with trends reported for each marker in the literature. Dots represent the average of two biological samples with two technical replicates each. **d** Personal pathway index (PPi) for each time point and the estimated differential PPi (dPPi) were calculated as explained in the text. P1: interleukin-12-mediated signalling pathway; P2: endodermal cell differentiation; P3: proteoglycan binding; P4: peroxidase activity; P5: complement cascade; P6: humoral immune response; P7: oncogenic MAPK signalling. **e** Distribution and density variation protein LFQ values (including two biological replicates) at each timepoint for each pathway assessed. Continuous bar in the middle represents the median. The thin discontinuous line represents the quartile lines. **f** Hierarchical clustering of the four urinary proteomes of Patient III. The protein LFQ values were used to perform the cluster analysis (with average linkage, no constraint, preprocessing with k-means and Euclidean distance).

**Table 2 The dPPi index column shows the index calculated as explained in the methods section.**

| Patient | Time point | dPPi index | dPPi Intervention? | Medical procedure | Outcome | dPPi intervention true? |
|---|---|---|---|---|---|---|
| 1 | 2 | 20 | Yes | Cytoscopy | Cystitis | Yes |
| 1 | 3 | −80 | No | Medical Monitoring* | No tumour | Yes |
| 1 | 4 | −61 | No | Medical Monitoring* | No tumour | Yes |
| 2 | 2 | 99 | Yes | Cytoscopy/RE-TURBT | Tis stage recurrence | Yes |
| 2 | 3 | 35 | Yes | Cytoscopy/TURBT | T1 stage recurrence | Yes |
| 2 | 4 | 5 | Yes | Cytoscopy | Recurrence (no AP exam) | Yes |
| 2 | 5 | 95 | Yes | Medical Monitoring* | Recurrence (no AP exam) | Yes |
| 3 | 2 | 36 | Yes | Cytoscopy/RE-TURBT | T2 Stage recurrence | Yes |
| 3 | 3 | −63 | No | Medical Monitoring* | No recurrence | Yes |
| 3 | 4 | −79 | No | Medical Monitoring* | No Recurrence | Yes |
| 4 | 2 | −80 | No | Medical Monitoring* | No Recurrence | Yes |
| 4 | 3 | −56 | No | Medical Monitoring* | No Recurrence | Yes |
| 5 | 2 | −3 | Yes | Cytoscopy/RE-TURBT | Ta stage recurrence | Yes |
| 5 | 3 | −2 | Yes | Medical Monitoring* | Nonneoplastic Bladder Mass | Yes |
| 6 | 2 | −41 | No | Cytoscopy | No recurrence | Yes |
| 6 | 3 | −7 | No | Medical Monitoring* | No Recurrence | Yes |

The dPPi intervention column shows whether medical intervention is necessary or not (Yes/No). The medical procedure column indicates the medical intervention performed for the patient, while the outcome column highlights the diagnosis given by the physicians. The column named 'dPPi intervention true?' confirms whether the intervention that was recommended by the dPPi index was necessary or not (Yes/No). Note that patient numbers 4, 5 and 6 are described in the supplementary Fig. 3, 4, and 5.

proteome was analysed again eight and eleven months after cystectomy, as shown in Fig. 4d timepoints 3 and 4, respectively. The expression levels of all the pathways were found to be lower after resection of the bladder (Fig. 4d). The dPPi decreased 63 and 79% between onset and time points three and four, respectively. The latter is consistent with the recovery experienced by this patient. Currently, this patient is in good health. The profile of the four proteomes since the onset of disease were compared (Fig. 4f) and clearly show that the urines were clustered into two groups (before and after radical cystectomy).

Three further patient examples are presented in the Supplementary Fig. 3, 4 and 5.

Table 2 shows the dPPi index for each time point for each patient in a comprehensive manner. First, the urine was monitored, the dPPi was calculated, and an alert for intervention was set as yes or no. Next, the patient was examined by the physician's team through a medical procedure as described in the table. The diagnostic provided was then compared with the alert provided by the dPPi. As shown in Table 2, 100% of the alerts (yes or no) were accurate.

In this work, we demonstrated how using the dPPi algorithm to assess the urine proteome is a robust tool to (i) monitor the course of BC disease and (ii) guide clinical decisions regarding intervention. As the disease progresses, the levels of selected cancer hallmark pathways change. This variation is used to monitor the course of BC disease in a simple, straightforward, and robust manner. This study would benefit from a large cohort of patients and will need to be replicated on a larger scale to validate the reported conclusions.

### Data availability

The mass spectrometry proteomics data that support the findings of this study have been deposited in ProteomeXchange Consortium[16] via the PRIDE[17] partner with the PXD025139 accession codes. Source data for the Figs. 1–4 and Supplementary Fig. 2–5 presented in the manuscript are available in the Supplementary Data 2.

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

## Acknowledgements

PROTEOMASS Scientific Society is acknowledged by the funding provided to the Laboratory for Biological Mass Spectrometry Isabel Moura (#PM001/2019 and #PM003/2016). Authors acknowledge the funding provided by the Associate Laboratory for Green Chemistry LAQV which is financed by national funds from FCT/MCTES, *Fundação para a Ciência e a Tecnologia* and *Ministério da Ciência, Tecnologia e Ensino Superior*, through the projects UIDB/50006/2020 and UIDP/50006/2020. H.M.S. acknowledges the Associate Laboratory for Green Chemistry-LAQV (LA/P/0008/2020) funded by FCT/MCTES for his research contract. L.B.C. is funded by the FCT/MCTES PhD grant 2019 (SFRH/BD/144222/2019). G.M. is funded by the FCT/MCTES PhD grant 2018 (SFRH/BD/139384/2018). H.López-Fernández is supported by a 'María Zambrano' post-doctoral contract from Ministerio de Universidades (Gobierno de España).

## Author contributions

J.L.C., C.L., L.C.P., and H.M.S. designed the experimental work and provide financial support. L.B.C. and G.M. performed the laboratory work and data analysis under the supervision of H.M.S. and J.L.C. H.M.S and J.L.C. drafted the manuscript. L.B.C., J.L.C., C.L., L.C.P, R.D., M.M. and H.M.S. revisited the drafted version, corrected it, and made valuable suggestions. H.L.F. and F.D. performed the statistical analysis. L.C.P. and M.M. managed patient interventions and provided samples and medical data.

## Competing interests

The authors declare no competing interests.
