## [Peer Review File · Communications Medicine]

Reviewers' comments:

Reviewer #1 (Remarks to the Author):

It is an interesting concept to use label-free proteomics to measure multiple biomarkers in patient specimens over time to monitor disease. However, the manuscript is not written in such a way that the reader can easily follow how the PPI is generated. My understanding after reading the manuscript is that the list of proteins in the PPI is based on comparing the urinary proteomes of a group of G3T1 patients with good outcomes versus a group of G3T1 patients with bad outcomes? Therefore, the list of proteins is not personal, just the changes relative to the index sample. If the PPI was generated by comparing 2 times per patient please make this clear. The manuscript could be suitable for publication if it was written more clearly. Perhaps a flow diagram of the PPI work-flow would help and also condensing the very complex figures to key points and putting the rest of the information in a supplement.

Specific points.

lines 38-43... there are four examples not three? Also, the two groups and what they are used for is not clear.

Figure 1 ... which proteins are shown in Figure 1a - is this just the proteins that differ significantly between the 2 groups? Why is which group which sample is in not shown? Some basic proteomic data summaries should be presented such as how many proteins are detected in each urine sample.

It would be better if more than 4 patients were followed longitudinally.

Reviewer #2 (Remarks to the Author):

The manuscript entitled "Pathway centered analysis to guide clinical decision-making in precision medicine" by Carvalho et al. proposed a new strategy of "diagnostics" based on mass spectrometry-based proteomics analysis. The concept is very interesting, and the manuscript is well laid out. However, the reviewer thinks that additional experiments and clarifications followed by major revision are required for the authors' conclusion.

Major Compulsory Revisions

1. I do not see the introduction. Therefore, I do not understand what the background is, including what we know and what we don't know, and what is (or are) the current challenge(s) that the authors try to address.
2. In Fig 1D, it is unclear why authors analyzed MPO? Fig 1D is not discussed in the manuscript.
3. I assume this study is discovery study. If so, cohort of 17 subject (?) is quite small. On page 4, it says that n = 71 (I assume this is typo). If this is correct, please provide the table for demographics. This is a good start, and this may be a good preliminary data for grant application, but I'm not sure for manuscript. It is also unclear who are the group A and group I. Perhaps, the subjects with deceased are group A and alive are group I. If this is correct, please specify in the table 1.

4. There are a lot of known protein markers for bladder cancer. In Fig 2D, why the authors selected the six proteins?
5. Fig 2 B-D and F-G are not discussed in the manuscript. Same in Fig 3 and 4.

Minor essential revisions

1. I am aware that this journal doesn't specify the format of the manuscript. Therefore, it is ok that the manuscript doesn't have the structure of introduction, results, and discussion. However, the minimal information should be included. The authors may want to rearrange the manuscript to provide the information.

Based on the cohort that the patients are mainly T1 (one patient is T2), I assume that the authors want to identify the patients who show worse prognosis. If we accurately identify the patients, the physicians can provide aggressive treatment to the patients who show worse prognosis, while can avoid aggressive treatment to the patients who show good prognosis. It is unfortunately unclear what is "clinical decision-making" that the authors are talking about, because the background of this study is not described in the manuscript. Authors may want to 1) discuss what is current AUA guideline for "Diagnosis and Treatment of Non-Muscle Invasive Bladder Cancer", 2) clarify what is the problem of the guideline, 3) compare the dPPI with the guideline, and 4) clarify the advantage of dPPI. Specifically, how the information based on dPPI changes the clinical decision-making? Decision between BCG/Maintenance and Chemotherapy/BCG Combinations? How does the decision improve clinical outcomes?

Reviewer #3 (Remarks to the Author):

The authors present a prospective evaluation of the urine proteome of patients with non-muscle invasive bladder cancer. this is a critical need, especially for T1 patients to identify those that will recur or progress. For the patient, any real biomarker that can aid in treatment decision would be a welcome addition to our current clinical and pathological staging. Thus, I applaud the overall strategy of the submission. However, several features could be better described

- 1) how many patients were included? The supp table 1 includes this data, but it is not in the MS
- 2) the figures show only anecdotes of a few patients. The authors state that decreases in PPI index are associated with improved disease outcomes, but how accurate is that finding?
- 3) it's hard to integrate all of the pathways. This seems very personalized. Decreases in "pro-oncogenes" pathways would be good, but generally increases in inflammation would be good. How was this interpreted?
- 4) why were no patients treated with intravesical therapy, such as BCG?

Reviewer #4 (Remarks to the Author):

Mass spectrometry-based proteomics has enabled the detection of biochemical changes at the protein level in an individual's proteome over time. This has provided the foundation for

personalized medicine initiatives. Toward this goal, Carvalho et al. are proposing a new concept named differential Personal Pathway Index (dPPI). In this study, the authors assessed the urinary proteomes of patients with bladder cancer using the dPPI algorithm in an effort to monitor the course of disease progression and guide clinical decision-making. Although the dPPI algorithm is novel, the concept of longitudinal profiling of individual proteomes is not new. For a few examples of other similar studies, the authors are encouraged to refer to PMID: 29361466, 31068711, 31767621, 32849183, and 31163045. The fundamental concern with this manuscript is that the dPPI algorithm is not presented in the context of other longitudinal proteome profiling studies. Accordingly, the title of the manuscript should be changed to reflect the study's focus on applying label-free quantitative proteomics to monitor the disease course of bladder cancer.

Major concerns

1. Line 28-30: No information is provided regarding the specific method of label-free quantitation (LFQ) upon which the dPPI algorithm is based: Peak area? Peak height? Spectral counting?
2. Line 33: "Based on variation in the levels of tens of signature proteins..." How was the optimal number of signature proteins determined? What is the specific number of signature proteins?
3. Suppl. Table 1: The significance of each of the drug treatments should be described. For the purpose of data interpretation, can the patients be grouped based on similar classes of prescribed medication?
4. Line 68: "This prompted us to call the patient for follow-up." The unstated implication is that the dPPI algorithm is being used for clinical decision-making; however, the authors do not provide any information regarding the clinical validation of the algorithm. This information is required to support the authors' claims that the dPPI algorithm has utility beyond a research environment.
5. Figure 2D and Lines 77-78: An explanation should be provided regarding why the trends in the levels of AGT and ORM1 do not follow the trends that are reported in the literature.
6. Figure 2F & 4F: It is unclear whether any of the protein LFQ values exhibit statistically significant changes among the sampling points.
7. Line 177: "Urine samples were collected...". Were the urine samples from random collections for first-morning voids? The difference between these collection methods has implications related to the concentrations of analytes in the urine.
8. Bioinformatics section: No information is provided regarding normalization of the mass spectrometry data.

Minor concerns

1. Throughout the manuscript the authors use the phrase, "biopsies, such as plasma, serum or saliva". Consider using the term "liquid biopsies" instead given that "biopsies" commonly are defined as cells or tissue, not fluids.
2. Suppl. Table 1: Specify the units for "Time of Recurrence", "Time of Progression", and "Global Survival".
3. Lines 204-205: The information in these two sentences appears to be redundant: "Peptides were resuspended in 100 μ L of 3% (v/v) acetonitrile containing 0.1% (v/v) aqueous formic acid (FA). Protein digests were resuspended in 100 μ L of 3% (v/v) acetonitrile containing 0.1% (v/v) aqueous formic acid (FA)...."

4. Methods – Patients: The authors should mention that the demographic information for the patients included in the study is included in Supplemental Table 1.

Answers to reviewer comments for manuscript entitled:

“Personal Pathway centered analysis to monitor disease course in precision medicine”
COMMSMED-21-0216

by Hugo Santos et al.

We indeed thanks all the constructive comments done by the reviewers. Truly appreciated indeed

Referee #1: Bladder cancer biomarkers, urine proteomics

Referee #2: Bladder cancer biomarkers, urine proteomics

Referee #3: Urology, bladder cancer (clinical)

Referee #4: Mass spectrometry, proteomics

Reviewers' comments:

Reviewer #1 (Remarks to the Author):

It is an interesting concept to use label-free proteomics to measure multiple biomarkers in patient specimens over time to monitor disease. However, the manuscript is not written in such a way that the reader can easily follow how the PPI is generated. My understanding after reading the manuscript is that the list of proteins in the PPI is based on comparing the urinary proteomes of a group of G3T1 patients with good outcomes versus a group of G3T1 patients with bad outcomes? Therefore, the list of proteins is not personal, just the changes relative to the index sample. If the PPI was generated by comparing 2 times per patient please make this clear. The manuscript could be suitable for publication if it was written more clearly. Perhaps a flow diagram of the PPI work-flow would help and also condensing the very complex figures to key points and putting the rest of the information in a supplement.

Answer to general comments

We thank the positive and constructive comments made by the referee. The PPI is generated with a set of proteins found differentially expressed between two different groups of patients. First, proteins differentially expressed between the two groups are found. Then from these groups of proteins the ones found systematically augmented in the group of patients with recurrence and/or progression were selected. We hypothesised that from the onset of BC the levels of such proteins could be used to follow the evolution of BC at an individual level. Our following-up in time of patients individually confirms our hypothesis. At each sampling point, the personal pathway index is calculated using this second group of proteins, which are then grouped by pathways of interest. The differential pathway index is calculated by subtracting two consecutive PPIs. If the difference is positive the patient is not progressing well.

The idea of making a workflow chart is excellent and we presented it in Fig. SM1.

As for the point about the figures, we consider the figures complex, yes, as the referee suggests. However, they are explained in figure captions in detail. Also, we consider it intuitive for non-experts in the area. Said this, if considering the new version of the manuscript, the referee considers it imperative to condense the figures, we are willing to remake them.

Specific points.

lines 38-43... there are four examples not three? Also, the two groups and what they are used for is not clear.

To make this point clearer, we have augmented the number of patients from 16 to 30. We have also divided the two groups of patients following a rationale medical outcome. The one group is composed of T1 patients that have experienced recurrence and/or progression, and the second group is composed of T1 patients that have not experienced it.

Finally, the number of patients followed with time has passed from 3 to 6.

Figure 1 ... which proteins are shown in Figure 1a - is this just the proteins that differ significantly between the 2 groups? Why is which group which sample is in not shown?

Yes, these are the proteins that differ significantly between the two groups and are the proteins used to select the oncogenic pathways. Then, the proteins belonging to each pathway are determined subsequently as is explained in Fig. SM1. Some basic proteomic data summaries should be presented such as how many proteins are detected in each urine sample.

We thank this excellent suggestion. The numbers of proteins detected are now presented in Fig. SM2

It would be better if more than 4 patients were followed longitudinally.

Following the advice of the reviewer, we have been able to increase de number of patients from 3 to 6. Also, we have augmented the time points for the original 3 patients.

Reviewer #2 (Remarks to the Author):

The manuscript entitled "Pathway centered analysis to guide clinical decision-making in precision medicine" by Carvalho et al. proposed a new strategy of "diagnostics" based on mass spectrometry-based proteomics analysis. The concept is very interesting, and the manuscript is well laid out. However, the reviewer thinks that additional experiments and clarifications followed by major revision are required for the authors' conclusion.

Major Compulsory Revisions

1. I do not see the introduction. Therefore, I do not understand what the background is, including what we know and what we don't know, and what is (or are) the current challenge(s) that the authors try to address.

We thank this observation. According to the reviewer's suggestion, we have remade the introduction to briefly address the problem and link it with our strategy to solve it.

2. In Fig 1D, it is unclear why authors analysed MPO? Fig 1D is not discussed in the manuscript.

The use of MPO protein is explained in figure 1 caption where we note: “D, Validation of mass spectrometry data is done using myeloperoxidase (MPO) protein label-free quantification (LFQ) and ELISA (ng/mL). MPO protein has been proposed recently as a marker of poor prognosis in lung3 and ovarian4 carcinomas. To the best of our knowledge, there are no studies reporting overexpression of MPO as a prognosis tool for BC carcinoma.

However, as it seems unclear the primary reason why we chose MPO, we have modified the text as follows: “The protein myeloperoxidase was found the one with the largest overexpression in patients with recurrence and /or progression. Therefore, was used it for quality control to validate the mass spectrometry data using Elisa as an orthogonal validation method (Fig. 1E and Fig. 1F). In addition, MPO protein has been proposed recently as a marker of poor prognosis in lung3 and ovarian4 carcinomas.”

3. I assume this study is discovery study. If so, cohort of 17 subject (?) is quite small. On page 4, it says that n = 71 (I assume this is typo). If this is correct, please provide the table for demographics. This is a good start, and this may be a good preliminary data for grant application, but I’m not sure for manuscript. It is also unclear who are the group A and group I. Perhaps, the subjects with deceased are group A and alive are group I. If this is correct, please specify in the table 1.

Following this comment, we have increased the number of patients to 30 and we have grouped them into two groups following the criteria of those that had experienced relapse and/or recurrence (group 1) versus those that had no experience relapse and/or recurrence (group 2).

This is not a discovery study as we are not using biomarkers but pathways. This concept is new.

4. There are a lot of known protein markers for bladder cancer. In Fig 2D, why the authors selected the six proteins?

This is an interesting question, and we thank this reviewer for making it. These proteins have been reported in literature as valuable biomarker candidates for bladder cancer. The corresponding references are cited in figure captions. For instance, high levels of APO4 and APOE have been described as markers of cancer presence. MMP9 is a marker of inflammation and invasiveness. They were also chosen because they perfectly reflect that a single biomarker is not a good option to follow an individual as some phenotypes do not express the trend which is expected for a given biomarker. This can be seen in some of our patients, thus when the trend is not the expected one, a red square is placed.

5. Fig 2 B-D and F-G are not discussed in the manuscript. Same in Fig 3 and 4.

Indeed truth. This is because they are briefly commented in figure captions. Figures showing volcano plots give an idea to mass spec experts and to readers about the differences found between the urine proteomes. The same applies to Fig. F, which is explained in figure captions. A detailed explanation in the text would be advisable, we agree. However, the limit in the number of words in the main text hampers us from doing so. Also, if the reviewer considers it necessary, they can be removed.

Minor essential revisions

1. I am aware that this journal doesn’t specify the format of the manuscript. Therefore, it is ok that the manuscript doesn’t have the structure of introduction, results, and discussion. However, the minimal information should be included. The authors may want to rearrange the manuscript to provide the information.

Thanks for this comment. Following this suggestion, we have now implemented sections in the manuscript according to the Journal guidelines.

Based on the cohort that the patients are mainly T1 (one patient is T2), I assume that the authors want to identify the patients who show worse prognosis. If we accurately identify the patients, the physicians can provide aggressive treatment to the patients who show worse prognosis, while can avoid aggressive treatment to the patients who show good prognosis. It is unfortunately unclear what is “clinical decision-making” that the authors are talking about, because the background of this study is not described in the manuscript. Authors may want to 1) discuss what is current AUA guideline for “Diagnosis and Treatment of Non-Muscle Invasive Bladder Cancer”, 2) clarify what is the problem of the guideline, 3) compare the dPPI with the guideline, and 4) clarify the advantage of dPPI. Specifically, how the information based on dPPI changes the clinical decision-making? Decision between BCG/Maintenance and Chemotherapy/BCG Combinations? How does the decision improve clinical outcomes?

Certainly, this comment is very welcome because it makes us realize that we were not clear in presenting our work. Therefore, we have changed the introduction accordingly to this new one:

“Thus, based on variation in the levels of tens of signature proteins, the dPPI will reflect both the response of the individual’s proteome to whatever course the disease takes and any medical care that ensues. The variation reflects the status of the biochemical pathways to which the proteins correspond, specifically chosen to combine multiple sources of diagnostic information. Thus, the evolution of the disease is monitored in a comprehensive yet simple manner. Essentially, the more positive the dPPI, the worse the course of disease; the more negative the dPPI, the better the clinical outlook. The dPPI can therefore flag the need for intervention when a disease is progressing. This concept is explained in this work with six examples of urinary proteomes of patients with bladder cancer (BC) that were followed up to 40 months.”

Our approach works as an early alert that cancer is recurring and or progressing. In other words, allows an easy follow of the course of the disease. No need for a cystoscopy intervention unless the alert is given. Also, it can flag that a treatment is not working.

Reviewer #3 (Remarks to the Author):

The authors present a prospective evaluation of the urine proteome of patients with non-muscle invasive bladder cancer. this is a critical need, especially for T1 patients to identify those that will recur or progress. For the patient, any real biomarker that can aid in treatment decision would be a welcome addition to our current clinical and pathological staging. Thus, I applaud the overall strategy of the submission. However, several features could be better described

We thank reviewer 3 for this motivating comments.

1)how many patients were included? The supp table 1, includes this data, but it is not in the MS

The number of patients has been increased as follows:

- (a) For selecting the proteins differentially expressed between groups, we have passed from 17 to 30 individuals. Now patients have been grouped into (i) patients experiencing recurrence and/or progression and (ii) patients that have not experienced recurrence and/or progression.

- (b) For exemplifying how to use the dPPI approach we have increased the number of patients from 3 to 6 and the number of sampling points in the first 3 patients.

The clinical and medical data about the patients is now presented in Figure 1.

2) the figures show only anecdotes of a few patients. The authors state that decreases in PPI index are associated with improved disease outcomes, but how accurate is that finding?

The number of patients followed now is six, including patients with different outcomes. Precisely, it is shown in the text how the dPPI can be interpreted to anticipate via urine proteome whether the cancer is recurring or progressing again, with no need for cystoscopy. Please note that the analysis allows not only to follow the selected pathways but also single biomarkers that can also provide extra medical information to confirm the alert arising by the dPPI approach.

3) it's hard to integrate all of the pathways. This seems very personalized. Decreases in "pro-oncogenes" pathways would be good, but generally increases in inflammation would be good. How was this interpreted?

This comment is quite interesting from a medical point of view. We do not want to open a debate that still is under progress in the medical community about whether inflammation processes are good or are not for the course of a disease. However, it seems that an agreement is being taken by the medical community in the sense that inflammation is a good process to some point when the benefits for the patient are no longer sustained due to the side effects of the inflammation response.

Following the referee's suggestion, we no longer take into consideration pathway 4 "Hydrogen peroxide catabolic process". This pathway is related to inflammation processes and was one of the pathways experiencing notable changes. It is worth noting that despite this the trend observed for the dPPI for every single patient is the same.

4) why were no patients treated with intravesical therapy, such as BCG?

We thank this interesting comment. Following the referee's suggestion, we have now included in Fig. 1B the medical treatment given to each patient. We have developed a concept that follows the course of disease for any patient in a truly personalised way. So far, the focus is not the treatment itself, but the course of the individual response.

Reviewer #4 (Remarks to the Author):

Mass spectrometry-based proteomics has enabled the detection of biochemical changes at the protein level in an individual's proteome over time. This has provided the foundation for personalized medicine initiatives. Toward this goal, Carvalho et al. are proposing a new concept named differential Personal Pathway Index (dPPI). In this study, the authors assessed the urinary proteomes of patients with bladder cancer using the dPPI algorithm in an effort to monitor the course of disease progression and guide clinical decision-making. Although the dPPI algorithm is novel, the concept of longitudinal profiling of individual proteomes is not new. For a few examples of other similar studies, the authors are encouraged to refer to PMID: 29361466, 31068711, 31767621, 32849183, and 31163045. The fundamental concern with this manuscript is that the dPPI algorithm is not presented in the context of other longitudinal proteome profiling studies. Accordingly, the title of the manuscript should be changed to reflect the study's focus on applying label-free quantitative proteomics to monitor the disease course of bladder cancer.

We thank the reviewer for these encouraging comments. Following the recommendation, we have now included the suggested examples in the text and references: 29361466, 31068711 and 32849183.

Dear editor and dear referee, we have mixed feelings about the suggestion regarding the title. On the one hand, the changes make sense as we have used mass spectrometry to follow bladder cancer. On the other hand, the dPPI concept is new and it can be extended to other diseases. Therefore, we propose the referee and editor the following change for the title:

Personal Pathway centered analysis to monitor disease course in precision medicine

We also comment that bladder cancer is used to exemplify the dPPI concept. We have also added the comment of the reviewer regarding novelty as follows: "The concept of longitudinal profiling of individual proteomes has been described in literature 3–5 but up to date, however, there is no easy way to make this in a simple, straightforward and robust manner."

Major concerns

1. Line 28-30: No information is provided regarding the specific method of label-free quantitation (LFQ) upon which the dPPI algorithm is based: Peak area? Peak height? Spectral counting? Precursor signal intensity

Thanks, this is an excellent alert. We have used precursors signal intensity. This is now explained in methods as follows: "Relative label-free quantification was performed using the precursor signal intensity method and carried out using a delayed normalization, MaxLFQ, on MaxQuant software V2.0.3.0"

2. Line 33: "Based on variation in the levels of tens of signature proteins..." How was the optimal number of signature proteins determined? What is the specific number of signature proteins?

Thanks again for this observation. To make this clear we have now modified Fig.1 and we have made the rationale for grouping patients clearer. Patients have been grouped into those having experienced recurrence and/or progression and those that have not. Also, the way the proteins were selected has been explained better in the main body text, whilst a comprehensive chart has been added in supplementary material, Fig.SM1.

3. Suppl. Table 1: The significance of each of the drug treatments should be described. For the purpose of data interpretation, can the patients be grouped based on similar classes of prescribed medication?

The dPPI concept has been designed to be applied to each patient individually. Therefore, the significance of each drug treatment makes sense only from this point of view. As the dPPI analyse the course of the disease using the proteome, the effects of the drugs are also reflected in the proteome. A detailed proteome to drug response is out of the scope of this manuscript. Also, our text is limited to 5000 words, which hamper us from dealing with many avenues that our approach opens in medicine.

4. Line 68: "This prompted us to call the patient for follow-up." The unstated implication is that the dPPI algorithm is being used for clinical decision-making; however, the authors do not provide any information regarding the clinical validation of the algorithm. This information is required to support the authors' claims that the dPPI algorithm has utility beyond a research environment.

We thank this comment. Very appropriated. We have reorganized the text, so now it is clearer (on our humble opinion) how the dPPI can be used to monitor and alert physicians of worsening during the disease course. For instance, when the MAPK pathway increases, when the dPPI increases, and when the individual biomarkers also increase, this is a sign of alert that something is going wrong and so to call the patient to perform a new urine analysis or a cystoscopy shall be the next step.

5. Figure 2D and Lines 77-78: An explanation should be provided regarding why the trends in the levels of AGT and ORM1 do not follow the trends that are reported in the literature.

We thank the reviewer for arising this time point. Biomarker response is phenotypically dependent and does not work for all patients in the same way. This question is being debated in the proteomics and medical community. This is, to which point biomarkers are reliable for 100% of patients?. In our examples, the biomarkers work well to form some patients. Precisely, one of the advantages of the dPPI approach is that relies on pathway variation instead of single protein variation.

6. Figure 2F & 4F: It is unclear whether any of the protein LFQ values exhibit statistically significant changes among the sampling points.

This point is very interesting. Following the referee's, suggestion, now we have marker the variations when they are statistically significant. However, whilst statistical significance is important when one single protein is used to set a difference, here the protein biomarkers presented are shown to display trends. For instance, the patient, I APOA4 presents a clear trend to lower values from point 1 to point 3. This trend changes in point 4. In the same way, CDH1 is presenting a trend towards higher values from point 1 to point 3, this trend changes in point 4.

7. Line 177: "Urine samples were collected...". Were the urine samples from random collections for first-morning voids? The difference between these collection methods has implications related to the concentrations of analytes in the urine.

Patients were instructed about how to collect second void morning urine.

8. Bioinformatics section: No information is provided regarding normalization of the mass spectrometry data.

Indeed, the referee is right. We have now answered these questions with a new section in supplementary material entitled "Bioinformatics section". Answering the reviewer question, in brief, basically, normalization was done Relative label-free quantification was performed using the precursor signal intensity method and carried out using a delayed normalization, MaxLFQ, on MaxQuant software V2.0.3.0. This is now included in the methods.

Minor concerns

1. Throughout the manuscript the authors use the phrase, "biopsies, such as plasma, serum or saliva". Consider using the term "liquid biopsies" instead given that "biopsies" commonly are defined as cells or tissue, not fluids.

We agree with this comment. Now the term liquid biopsy is used throughout the manuscript.

2. Suppl. Table 1: Specify the units for "Time of Recurrence", "Time of Progression", and "Global Survival".

Thanks for this note. We have now specified the units.

3. Lines 204-205: The information in these two sentences appears to be redundant: “Peptides were resuspended in 100 μ L of 3% (v/v) acetonitrile containing 0.1% (v/v) aqueous formic acid (FA). Protein digests were resuspended in 100 μ L of 3% (v/v) acetonitrile containing 0.1% (v/v) aqueous formic acid (FA)....”

Thanks, we have corrected it accordingly in Fig.1B

4. Methods – Patients: The authors should mention that the demographic information for the patients included in the study is included in Supplemental Table 1.

Thanks, now demographic information is included in Fig. 1 .

Reviewers' comments:

Reviewer #1 (Remarks to the Author):

Whilst the study is improved by increasing the number of patients it is still not written in a way which makes it completely clear what was done! In addition, I remain unconvinced that there are any longitudinal trends in the data that predict future recurrence or progression.

In the abstract background do you mean "anticipate" or "detect"? The comparator patient groups are not clearly described - did they have a tumour at the time when the urine was collected.

In the plain language summary you mention "tens of hundreds of protein" but the method actually only identified 300-400 proteins in each sample.

You added 38 mg boric acid to the samples. Does this lyse the cancer cells and release their contents?

Why do you state 90-100 minute gradient? Surely constant retention times are needed?

If 300-400 proteins were identified per sample then it is surprising that 197 are differentially expressed, especially as the identified proteins are mostly abundant plasma proteins (always a problem with urine proteomics) - does the data really contain useful data on intracellular signaling? How does haematuria influence the results?

How are p-values calculated for the volcano plots for individual patients as only duplicate measurements were taken?

None of the PPI data for patients 1-4 seems to show any substantial changes e.g. a rapid decrease would be expected after TURBT.

Do the authors have any biological rationale for their results? Do they think the proteomic changes in the urine reflect a malignant field change, tumour characteristics or residual disease?

Reviewer #4 (Remarks to the Author):

The authors have carefully considered the concerns raised by the reviewers regarding the initially submitted version of the manuscript, and the manuscript has been substantially revised accordingly. There are a few lingering minor grammatical errors which should be corrected. Additionally, the authors are strongly urged to consider revising the ROC curves presented in Fig. 5B and 5C given that the way they are currently displayed (Sensitivity vs. Specificity) is unconventional; ROC curves are conventionally depicted as Sensitivity vs. 1-Specificity.

Reviewer #5 (Remarks to the Author):

The manuscript is interesting, however there are a few issues which need to be addressed. My review is mainly from a statistical point of view. Some comments:

1. Another reviewer has raised a question on the selection of biomarkers to consider. How was this selection made? It appears that the selected markers are supported by references but I suppose the question is why were some other markers not included?
2. How were patients selected? It seems that the proportion of patients who have had a recurrence is higher than the proportion that would have been expected in a random sample of such patients.
3. The number of patients is very small, but one can still do meaningful analyses. However such analyses are likely to only yield tentative conclusions. Doing analysis based on ROC curves is likely to lead to overfitting with no way to assess this. I suggest to remove this analysis and make any conclusions less strong, e.g. to frame it as an exploratory study which will need replication in a larger sample.
4. Power calculations after the data have been collected and analysis has been done are not useful and can be misleading.

Reviewer #6 (Remarks to the Author):

This is an article investigating the role of urine proteomes for bladder cancer surveillance. Most questions and comments were addressed adequately. My only suggestion is on the study title. The current title of the paper is far too general, and the readers will have no clue that this paper is about proteomes just by reading the title alone.

Answers to referee comments on manuscript COMMSMED-21-0216B (second revision)

Reviewers' comments and answers to:

Answers to comments raised by reviewer 1.

Reviewer #1 (Remarks to the Author):

Whilst the study is improved by increasing the number of patients it is still not written in a way which makes it completely clear what was done! In addition, I remain unconvinced that there are any longitudinal trends in the data that predict future recurrence or progression.

Answer to comment 1. Reviewer 1.

We thank the reviewer for raising this interesting point.

First, whilst we do believe our method can be used to follow BC disease course and help to flag intervention. However, following the referee's suggestion, we have removed any reference to predicting future recurrence or progression. In addition, we have replaced these expressions with the term "flag intervention".

Furthermore, we understand that a large cohort of patients would be desirable, and so we have noticed this in the final part of the manuscript as follows:

" This study would benefit from a large cohort of patients and will need replication on a larger scale to validate the conclusions here reported"

Second, following the referee's suggestion, we have rewritten the introduction to make it more transparent as follows:

Bladder cancer (BC) is among the worst neoplasms because of its high incidence and mortality and because cystoscopy is necessary to diagnose BC. If BC is confirmed, the

patient undergoes surgery, then the patient must undergo a control cystoscopy every three months. Cystoscopy is an expensive intervention that requires a surgical block and a minimum of two physicians and one nurse. However, an even worse problem is the detrimental effect on the patient's state of mind, as cystoscopy is highly invasive and is performed through the urethra. Furthermore, some patients experience a rapid recurrence; thus, the control intervention is performed too late, and transurethral resection of the bladder tumour is needed. Therefore, new approaches to diagnose BC and follow-up are needed.

The biochemical changes that occur at the protein level in an individual after they develop a disease provides information, and with this information, we can adjust therapy, provide follow up medical care and make precision medicine truly personalized. The concept of performing longitudinal profiling with individual proteomes has been described in the literature; however, to date, there is no easy way to achieve this in a simple, straightforward and robust manner. The possibility of following the course of BC using the urinary proteome has not yet been described in the literature. To overcome this gap, we explored the evolution of the urinary proteome of BC patients using high-resolution mass spectrometry-based proteomics in conjunction with advanced bioinformatics tools. Thus, we introduced a new concept named the differential personal pathway index (dPPI), which uses the variation in the expression of selected biochemical pathways, which is calculated using a large number of urine proteins that are linked to Hanahan and Weinberg's biological hallmarks of cancer (HWhc). Thus, based on variation in the levels of tens of signature urine proteins, the dPPI reflects both the response of the individual's urine proteome to whatever course the disease takes and any medical care that ensues. The variation reflects the status of the biochemical pathways to which the proteins correspond, and these pathways are specifically chosen to combine multiple sources of diagnostic information. Thus, in this work, we showed how to monitor the evolution of BC in a comprehensive yet straightforward and personalized manner

using the patient's urine proteome. Essentially, the more positive the dPPi is, the more severe the course of BC; the more negative the dPPi is, the better the clinical outlook. Therefore, the dPPi flags the need for medical intervention at an early stage. This work further demonstrates this concept using six patients diagnosed with T1-BC who were followed for 62 months.

In the abstract background, do you mean "anticipate" or "detect"?

Answer to comment 2. Reviewer 1.

We thank the reviewer for this interesting comment. As a matter of fact, the phrase was not clear enough, and we have remade it from this:

"The information encoded in the proteome phenotype of everyone contains the key to anticipate the outcome of a disease."

To this one

"The key to following the disease outcome is contained within the information encoded in the proteome phenotype of every individual.."

The comparator patient groups are not clearly described - did they have a tumour at the time when the urine was collected?

Answer to comment 3. Reviewer 1.

We thank the reviewer for this comment. The information requested is in Fig. 1. All patients were diagnosed with the T1-Cancer stage when the urine was collected before intervention.

In the plain language summary, you mention "tens of hundreds of proteins" but the method actually only identified 300-400 proteins in each sample.

Answer to comment 4. Reviewer 1.

This is an excellent point, certainly. We have replaced the phrase "to quantify tens of hundreds of proteins" with "to quantify hundreds of proteins"

You added 38 mg of boric acid to the samples. Does this lyse the cancer cells and release their contents?

Answer to comment 5. Reviewer 1.

Great point indeed. No, the boric acid does not lyse cancer cells, nor any other type of cells present in the urine. For instance, when we get urine with hematuria, and when centrifuge it, the red cells are eliminated as a precipitate, and the urine recovers its typical yellow colour. Something that would not happen if the red cells had released their content.

Further Info here:

DOI: [10.1128/CMR.00030-15](https://doi.org/10.1128/CMR.00030-15)

Why do you state 90-100 minute gradient? Surely constant retention times are needed?

Answer to comment 6. Reviewer 1.

Thanks for rising this point. What it is described is just the HPLC protocol. Giving a lecture to the complete one will, perhaps, make clear the point:

"Then the peptides were separated using an analytical column (Acclaim™ PepMap™ 100C18, 2 μm, 0.075 mm i.d x 150 mm) with a linear gradient at 300 nL.min⁻¹ (mobile phase A:

aqueous FA 0.1% (v/v); mobile phase B 90% (v/v) acetonitrile and 0.08% (v/v) FA) 5-90 min from 5% to 35% of mobile phase B, 90-100min linear gradient from 35% to 95% of mobile phase B, 100-110 min 95% B.”

If 300-400 proteins were identified per sample, then it is surprising that 197 are differentially expressed, especially as the identified proteins are mostly abundant plasma proteins (always a problem with urine proteomics).

Answer to comment 7. Reviewer 1.

Thanks for bringing up this interesting topic. The maximum number of proteins identified in one single urine of our set was 494. The number of unique proteins considering ALL the samples of our set was about 754. For our studies, we have chosen the proteins that are found common in at least 50% of the urines of each group. This way, the number of proteins used in Fig. 1 (recurrence versus no recurrence) was 380.

To show the correlation between the most abundant plasma proteins and the proteins we have found in urine, we have prepared the Figure embedded below. It may be seen that there are many proteins in the urine that do not match the concept of the most abundant plasma proteins. The following manuscript (<https://doi.org/10.1016/j.cels.2016.02.015>) was used as a reference to obtain the most abundant plasma proteins.

FigRev1- Venn Diagram comparing the most abundant proteins in plasma and urine. A.: (I) The number of proteins found in Plasma by high-resolution mass spectrometry as described in manuscript <https://doi.org/10.1016/j.cels.2016.02.015>. (II) The number of proteins found in the urine in our work. B.: (I) Diagram of 20 most abundant proteins found in plasma by high-resolution mass spectrometry as described in manuscript <https://doi.org/10.1016/j.cels.2016.02.015>. (II) 20-Most abundant proteins present in the plasma found in the urine in our work.

- does the data really contain useful data on intracellular signalling?

Answer to comment 8. Reviewer 1.

We thank this comment as it also opens new avenues for future work in our study. Because of apoptosis, eventually, some intracellular signalling proteins can be found. We conducted research on our dataset using the Gene Ontology Annotations GO:0035556 (intracellular signal transduction) database (FigRev2), and we have only found 2 related intracellular

signalling proteins, namely, Complement C3 and Prothrombin. However, the study of intracellular proteins is out of the scope of this manuscript.

FigRev2- Venn Diagram comparing the GO:0035556 (intracellular signal transduction) with the protein dataset of T1 bladder cancer patients used in our study.

- How does haematuria influence the results?

Answer to comment 9. Reviewer 1.

Thanks for this interesting question. Samples with hematuria are not included in our study. So far, we have added this information in the methods section and sample treatment subsection: "samples with hematuria were not included in our study."

How are p-values calculated for the volcano plots for individual patients as only duplicate measurements were taken?

Answer to comment 10. Reviewer 1.

Thanks for this question. p-values were calculated using Perseus software using the Student's t-test (FDR 0.05 and SO of 0.1). It was done just following the standard procedure.

None of the PPI data for patients 1-4 seems to show any substantial changes e.g. a rapid decrease would be expected after TURBT.

Answer to comment 11. Reviewer 1.

Thanks for this Question. The changes in the PPI values are enough to calculate a difference between time points and to express such difference as % (dPPI). The changes can be easily seen in the figures.

On the other hand, a rapid decrease of PPI values after TURBT depends on different factors: 1.- patient phenotype; 2.- the success of the medical intervention and 3.- time in which the urine is analyzed after medical intervention. Therefore, a rapid decrease will depend essentially on each patient.

Do the authors have any biological rationale for their results? Do they think the proteomic changes in the urine reflect a malignant field change, tumour characteristics or residual disease?

Answer to comment 12. Reviewer 1.

Thanks for this interesting comment. The bladder is in contact with the urine, which is reflected in the urine protein content of the patient. Thus, As the tumour rises and evolves, the proteins found in urine change; many of these changes are in protein levels, especially in

those related to cancer growth, inflammation, and immune response. This is what we follow using the urinary proteome so intervention or no intervention can be decided without cystoscopy.

Answers to comments raised by reviewer 4.

Reviewer #4 (Remarks to the Author):

The authors have carefully considered the concerns raised by the reviewers regarding the initially submitted version of the manuscript, and the manuscript has been substantially revised accordingly. There are a few lingering minor grammatical errors which should be corrected. Additionally, the authors are strongly urged to consider revising the ROC curves presented in Fig. 5B and 5C given that the way they are currently displayed (Sensitivity vs. Specificity) is unconventional; ROC curves are conventionally depicted as Sensitivity vs. 1-Specificity.

Answer to comment 1. Reviewer 4.

We thank this valuable comment of referee 4. Following editors' indications, the ROC curves have been deleted from the manuscript. And so, the section named "Statistical Assays" is no anymore on it.

As for the grammatical errors, we also acknowledged this comment. The manuscript has been sent for revision to <https://authorservices.springernature.com/>

Answers to comments raised by reviewer 5.

Reviewer #5 (Remarks to the Author):

The manuscript is interesting; however, there are a few issues which need to be addressed.

My review is mainly from a statistical point of view. Some comments:

1. Another reviewer has raised a question on the selection of biomarkers to consider. How was this selection made? It appears that the selected markers are supported by references but I suppose the question is why were some other markers not included?

Answer to comment 1. Reviewer 5.

Thanks for this comment. As this selection of proteins is very important, we have added an explanation of the procedure used to select them in the supplementary material sections as follows:

“First, the urine proteome of patients with recurrence was compared with that of patients with no recurrence. A set of 197 dysregulated proteins were found.

Second, the differentially expressed proteins were used to determine the most dysregulated pathways.

Third, these pathways with statistical relevance (using p-value) were assessed to find out the ones with most of their proteins presenting the same tendency with time (increasing or decreasing)

Four, the pathways selected, as explained in 3, were cross-checked against the Hanahan and Weinberg biological hallmarks of cancer. Then seven pathways were selected, focusing on inflammation and immunology responses.

So far, 35 proteins belonging to these seven pathways were used to calculate the differential personal pathway index, dPPi, as defined in (1). (Proteins listed in Table SM2).

2. How were patients selected? It seems that the proportion of patients who have had a

recurrence is higher than the proportion that would have been expected in a random sample of such patients.

Answer to comment 2. Reviewer 5.

Thanks for making this question. The presence or absence of recurrence was the main reason for selection as we were trying to find out the differences between patients with recurrence versus patients with no recurrence,

3. The number of patients is very small, but one can still do meaningful analyses. However, such analyses are likely to only yield tentative conclusions. Doing analysis based on ROC curves is likely to lead to overfitting with no way to assess this. I suggest to remove this analysis and make any conclusions less strong, e.g. to frame it as an exploratory study which will need replication in a larger sample.

Answer to comment 3. Reviewer 5.

Indeed, this is a valuable comment. Following this suggestion, the ROC curves have been eliminated from the manuscript. Also, the following phrase has been added to the text:

“This study would benefit from a large cohort of patients and will need to be replicated on a larger scale to validate the reported conclusions.”.

4. Power calculations after the data have been collected and analysis has been done are not useful and can be misleading.

Answer to comment 4. Reviewer 5.

Indeed, this is a comment that needs to be implemented in the manuscript, and as such, the power calculations have been deleted. (ROC curves and power have been deleted)

Answers to comments raised by reviewer 6.

Reviewer #6 (Remarks to the Author):

This is an article investigating the role of urine proteomes for bladder cancer surveillance. Most questions and comments were addressed adequately. My only suggestion is on the study title. The current title of the paper is far too general, and the readers will have no clue that this paper is about proteomes just by reading the title alone.

Answer to comment 1. Reviewer 6.

We thank this comment, and it is worth implementing. So far, now we propose the following title:

“Personal proteome pathway-centred analysis to monitor the disease course in patients with bladder cancer”.

REVIEWERS' COMMENTS:

Reviewer #1 (Remarks to the Author):

The authors have addressed many of my concerns although mostly only in their responses and not in the manuscript!

This text (for example) "The maximum number of proteins identified in one single urine of our set was 494. The number of unique proteins considering ALL the samples of our set was about 754. For our studies, we have chosen the proteins that are found common in at least 50% of the urines of each group. This way, the number of proteins used in Fig. 1 (recurrence versus no recurrence) was 380." should be in the manuscript.

The manuscript is written as if in-depth data on signaling pathways has been gathered (MAPK kinase signaling is mentioned 7 times) but this cannot be the case as in their responses the authors state that "we have only found 2 related intracellular signalling proteins, namely, Complement C3 and Prothrombin". The proteins supporting the pathways (presented in table SM2) are not that convincing. Overall there seems to be a disconnect between the relatively modest number of proteins identified and the pathways being quantitated. Perhaps if Table SM2 was in the main manuscript then this might be more apparent to readers that are not familiar with proteomic data.

Reviewer #5 (Remarks to the Author):

My previous comments have been adequately addressed.

Answers to referee comments on manuscript COMMSMED-21-0216C

REVIEWERS' COMMENTS:

Reviewer #1 (Remarks to the Author):

The authors have addressed many of my concerns although mostly only in their responses and not in the manuscript!

This text (for example) "The maximum number of proteins identified in one single urine of our set was 494. The number of unique proteins considering ALL the samples of our set was about 754. For our studies, we have chosen the proteins that are found common in at least 50% of the urines of each group. This way, the number of proteins used in Fig. 1 (recurrence versus no recurrence) was 380." should be in the manuscript.

R: We thank the reviewer for suggesting including our direct answers to him in the main body text. As the new text to be added is one short sentence, we have addressed this request in the new version (version 4th). The new phrase is included in red in the version with changes highlighted.

Legend of Figure 1: "The maximum number of proteins identified in one single urine of our set was 494. The number of unique proteins considering all the samples of our set was about 754. For our studies, we have chosen the proteins that are found common in at least 50% of the urines of each group. This way, the number of proteins used in Fig. 1 (recurrence versus no recurrence) was 380.

"

The manuscript is written as if in-depth data on signaling pathways has been gathered (MAPK kinase signaling is mentioned 7 times) but this cannot be the case as in their responses, the authors state that "we have only found 2 related intracellular.

signalling proteins, namely, Complement C3 and Prothrombin". The proteins supporting the pathways (presented in table SM2) are not that convincing. Overall there seems to be a disconnect between the relatively modest number of proteins identified and the pathways being quantitated. Perhaps if Table SM2 was in the main manuscript, then this might be more apparent to readers that are not familiar with proteomics data.

Answer: This is an excellent comment. We have now added Table SM2 to the text as Table 1.

Reviewer #5 (Remarks to the Author):

My previous comments have been adequately addressed.